# Long Noncoding RNA *PVT1* Is Regulated by Bromodomain Protein BRD4 in Multiple Myeloma and Is Associated with Disease Progression

**DOI:** 10.3390/ijms21197121

**Published:** 2020-09-27

**Authors:** Hiroshi Handa, Kazuki Honma, Tsukasa Oda, Nobuhiko Kobayashi, Yuko Kuroda, Kei Kimura-Masuda, Saki Watanabe, Rei Ishihara, Yuki Murakami, Yuta Masuda, Ken-ichi Tahara, Hisashi Takei, Tetsuhiro Kasamatsu, Takayuki Saitoh, Hirokazu Murakami

**Affiliations:** 1Department of Hematology, Gunma University Graduate School of Medicine, 3-39-22 Showa-machi, Maebashi, Gunma 371-8511, Japan; nobuhiko.kobayashi@gmail.com (N.K.); kenichit3@gmail.com (K.-i.T.); m14702054@gunma-u.ac.jp (H.T.); 2Department of Laboratory Science, Gunma University Graduate School of Health Science, 3-39-22 Showa-machi, Maebashi, Gunma 371-8511, Japan; m12203035@gunma-u.ac.jp (K.H.); m14711022@gunma-u.ac.jp (Y.K.); kei-masuda@gunma-u.ac.jp (K.K.-M.); m13203040@gunma-u.ac.jp (S.W.); m13203005@gunma-u.ac.jp (R.I.); m13203034@gunma-u.ac.jp (Y.M.); yuuken.0420@gmail.com (Y.M.); kasamatsu@gunma-u.ac.jp (T.K.); tsaitoh@gunma-u.ac.jp (T.S.); hmura@gunma-u.ac.jp (H.M.); 3Institute of Molecular and Cellular Regulation, Gunma University, 3-39-22 Showa-machi, Maebashi, Gunma 371-8511, Japan; toda@gunma-u.ac.jp

**Keywords:** long noncoding RNA, *PVT1*, *MYC*, bromodomain, multiple myeloma

## Abstract

Long noncoding RNAs (lncRNAs) are deregulated in human cancers and are associated with disease progression. *Plasmacytoma Variant Translocation 1* (*PVT1)*, a lncRNA, is located adjacent to the gene *MYC*, which has been linked to multiple myeloma (MM). *PVT1* is expressed in MM and is associated with carcinogenesis. However, its role and regulation remain uncertain. We examined *PVT1*/*MYC* expression using real-time PCR in plasma cells purified from 59 monoclonal gammopathy of undetermined significance (MGUS) and 140 MM patients. The MM cell lines KMS11, KMS12PE, OPM2, and RPMI8226 were treated with JQ1, an *MYC* super-enhancer inhibitor, or MYC inhibitor 10058-F4. The expression levels of *PVT1* and *MYC* were significantly higher in MM than in MGUS (*p* < 0.0001) and were positively correlated with disease progression (*r* = 0.394, *p* < 0.0001). JQ1 inhibited cell proliferation and decreased the expression levels of *MYC* and *PVT1*. However, 10054-F4 did not alter the expression level of *PVT1*. The positive correlation between *MYC* and *PVT1* in patients, the synchronous downregulation of *MYC* and *PVT1* by JQ1, and the lack of effect of the *MYC* inhibitor on *PVT1* expression suggest that the expression of these two genes is co-regulated by a super-enhancer. Cooperative effects between these two genes may contribute to MM pathogenesis and progression.

## 1. Introduction

Multiple myeloma (MM) is a plasma cell neoplasm characterized by the proliferation of atypical plasma cells in the bone marrow and the production of monoclonal immunoglobulins. MM progresses from a precancerous state called monoclonal gammopathy of undetermined significance (MGUS) at a rate of approximately 1% per year [1,2,3]. The primary molecular mechanism of MM development is thought to be the activation of cancer-related genes by translocations of immunoglobulin heavy chain genes (IgH). The effects of these translocations include increased cyclin D1 expression by the translocation t(11; 14) IgH-*CCND1*, increased *FGFR3/MMSET* expression by t(4; 14) IgH-*FGFR3/MMSET*, and increased *c-MAF* expression by t(14; 16) IgH-*c-MAF* [4]. These chromosomal abnormalities are observed at the MGUS stage, so additional abnormalities are required for progression to MM. Recent genomic and transcriptomic analyses have shown that oncogene mutations, such as *RAS* mutations, and aberrant overexpression of *MYC* play an important role in the progression of MM [5]. However, not all players have yet been elucidated.

Recent transcriptome-wide analyses have revealed a large number of noncoding RNAs that are transcribed but not translated and can influence a range of cellular processes, including cellular proliferation, apoptosis, and motility [6,7]. Among the noncoding RNAs, long noncoding RNAs (lncRNAs), transcripts >200 nucleotides in length, have emerged as a class of key regulatory RNAs [8]. LncRNAs are deregulated in many human cancers and are associated with disease progression [9,10,11]. Several studies, including ours, have shed light on the role of lncRNAs in MM progression [12,13,14].

*Plasmacytoma Variant Translocation 1* (*PVT1)* is a lncRNA longer than 500 nucleotides, first found in mouse plasmacytoma [15] and then reported to be involved in the oncogenesis of many types of cancers [16,17]. *PVT1* is located at the 8q24 locus adjacent to *MYC* [18], which is highly expressed in many types of cancer and plays an important role in carcinogenesis [19,20]. *PVT1* is elevated in MM [5,21] and coamplified with *MYC* in many cancers [18]; there is an association between *PVT1* expression level and poor prognosis in many cancers [16,17,22,23,24]. High-level amplification and/or overexpression of *PVT1* is associated with an invasive phenotype of breast cancer and reduced survival time in ovarian cancer patients [25]. These observations indicate the involvement of *PVT1* in the maintenance of a transformed phenotype. However, its regulation and clinical significance in MM are poorly documented.

Super-enhancers are areas of the genome at which mediator complexes, including activators and coactivators, accumulate at higher densities than on regular enhancers. Super-enhancers controlling the expression of genes involved in cell identity, determination, and disease were recently described [26]. The super-enhancers are bound by the bromodomain-containing protein 4 (*BRD4*), a member of the bromodomain protein family that is essential for RNA transcription and transcription elongation [27]. The most well-known mechanism of *MYC* overexpression is the fusion of the IgH enhancer and *MYC* produced by the chromosomal translocation t(8; 14) in Burkitt lymphoma. *MYC* transcription is controlled by a super-enhancer [27,28], and BRD4 inhibitors markedly decrease *MYC* expression in many types of cells, including MM cells [28]. It was speculated that *PVT1* is regulated by a super-enhancer.

To clarify the mechanism of regulation of *PVT1* expression and the relationship of PVT1 with progression and prognosis in MM, we investigated *PVT1* expression in plasma and MM cell lines focusing on a super-enhancer-related mechanism, and the correlation between *PVT1* and *MYC* expression in MM and MGUS patients.

## 2. Results

### 2.1. PVT1 and MYC Expression in Plasma Cells of MM Is Higher than in MGUS or Control

The expression levels of *PVT1* and *MYC* in plasma cells were significantly higher in MM (mean: *PVT1* 2.58, *MYC* 0.74) than in MGUS (mean: *PVT1* 0.88, *MYC* 0.06) and the control (mean: *PVT1* 0.06, *MYC* 0.07) (*p* < 0.001, *p* < 0.001, respectively; Figure 1A,B). *PVT1* expression seemed to increase with disease progression, but it did not differ between samples from different stages (stages are defined according to the international staging system (ISS) reflecting progression) (*p* = 0.145, Figure 1C). We then compared *PVT1* expression levels between cell lines with different chromosomal abnormalities, as detected using interphase fluorescence in situ hybridization (iFISH) analysis, including t(11; 14), t(4; 14), t(14; 16), deletion 13q, and deletion 17p, and found no differences (*p* = 0.509, Figure 1D). Since *PVT1* is located on chromosome 8q24 and co-occurrence of 8q24 abnormality is sometimes observed in MM, we compared the expression levels between cell lines with 8q24 abnormalities, including t(8;14), and tested for 8q24 amplification using FISH analysis. However, no differences were found (Figure 1E). When we analyzed *PVT1* and *MYC* expression levels in the same patients, a positive correlation was found in both MM and MGUS patients (*r* = 0.484, *p* < 0.001; *r* = 0.423, *p* < 0.0001; respectively; Figure 1E,F).

### 2.2. BRD4 Inhibitors Inhibit MM Cell Proliferation and Downregulate PVT1 and MYC Expression

To clarify the mechanisms regulating *PVT1* expression, we focused on *BRD4* because *MYC* expression is regulated by BRD4. BRD4 inhibitors inhibited the proliferation of eight MM cell lines. JQ1 inhibited the proliferation of the cell lines KMS11, KMS12PE, KMS12BM, KMS26, KMM1, OPM2, and RPMI8226 at a concentration of 1 μM (Figure 2A). CPI-203 inhibited the proliferation of KMS11, KMS12BM, KMS26, and OPM2 at a lower concentration (0.1 μM), and inhibited KMS12PE, KMS18, KMM1, and RPMI8226 at 1 μM concentration (Figure 2B).

JQ1 reduced the *PVT1* mRNA expression at a concentration of 1 μM in KMS11, OPM2, KMS12PE, KMS12BM, KMM1, RPMI8226, and KMS26 (Figure 2C). As with JQ1, another BRD4 inhibitor, CPI-203, reduced the *PVT1* mRNA expression at a concentration of 1 μM, in KMS11, OPM2, KMS12PE, KMM1, RPMI8226, and KMS26 (Figure 2D).

Consistent with previous reports, 1 μM of JQ1 and CPI-203 significantly reduced *MYC* mRNA expression in KMS11, OPM2, KMS12PE, KMS12BM, KMM1, RPMI8226, and KMS26 (Figure 2E,F).

### 2.3. MYC Inhibitor Did Not Reduce PVT1 Expression

From the results described above, there were two possible explanations for the high *PVT1* expression levels in MM: *MYC* and *PVT1* are co-regulated by BRD4 or *PVT1* is regulated by *MYC*, which in turn is regulated by BRD4. To test these hypotheses, we used the *MYC* inhibitor 10058-F4, which inhibits *MYC* transcription activity by dissociating the MYC-MAX transcription complex. *PVT1* expression was not significantly altered by treatment with 10058-F4 in KMS11, OPM2, KMS12PE, or RPMI8226 (Figure 3). Therefore, no contribution of *MYC* transcriptional activity to *PVT1* expression was observed.

### 2.4. PVT1 Downregulation by Locked Anti-Sense Nucleotide Reduced MYC Expression

To determine the role of *PVT1* on *MYC* expression in MM cells, antisense locked nucleotide GapmeR^TM^ was used to knockdown *PVT1* expression. Three sequences of GapmeR were constructed and five MM cell lines were tested: KMS11, KMS12PE, KMM1, OPM2, and RPMI8226. *PVT1* expression was successfully knocked down in two cell lines, KMS11 and OPM2, by one GapmeR product (Figure 4A). In these two cell lines, *MYC* mRNA expression was significantly reduced (Figure 4B).

### 2.5. Clinical Significance of PVT1 Expression in MM

To investigate the clinical significance of *PVT1* in MM, the overall survival (OS) and progression-free survival (PFS) of MM patients were analyzed by dividing the two groups according to *PVT1* expression levels using quartiles. Eighty-nine newly diagnosed multiple myeloma (NDMM) patients were analyzed for OS and 85 NDMM patients were analyzed for PFS. Seven patients were excluded from OS analysis and 11 patients were excluded from PFS analysis due to the lack of reliable clinical data. The patients with high *PVT1* expression tended to have a shorter OS, with a median of 2.7 years compared to 5.0 years in those with low expression (*p* = 0.077, Figure 5A). PFS also tended to be shorter in patients with high *PVT1* expression, but the difference did not reach statistical significance (*p* = 0.095, Figure 5B).

For further clarification of the clinical significance of *PVT1* expression, OS and PFS were analyzed separatory by subcategories, including patients eligible or ineligible for autologous stem cell transplantation (ASCT), harboring high-risk or standard-risk cytogenetics, and harboring deletion 17p (del 17p). The ASCT eligible patients with high *PVT1* expression showed significantly shorter OS (*p* = 0.004), but this significant difference was not found in the ASCT ineligible patients (*p* = 0.66). PFS in patients with high *PVT1* expression in both groups was not different. OS and PFS between patients with high and low *PVT1* expression did not differ in the groups of patients harboring high-risk and standard-risk cytogenetics. However, significantly shorter OS and PFS of the patients with high *PVT1* expression was found in the groups of patients who did not harbor del 17p (*p* = 0.008, *p* = 0.0043, respectively), although this significant difference was not observed in the patients with del 17p.

## 3. Discussion

In this study, we aimed to clarify the mechanism of regulation of *PVT1* expression and its relationship with progression and prognosis in MM. We found significantly higher *PVT1* and *MYC* expressions in MM and MGUS plasma cells compared to the control cell line, suggesting that the lncRNA *PVT1* is associated with MM pathogenesis and progression. A positive correlation between *MYC* and *PVT1* expression levels suggested that *MYC* and *PVT1* are co-regulated by the same mechanism. Two *BRD4* inhibitors, JQ1 and CPI -203, reduced both *MYC* and *PVT1* in MM cell lines, but the *MYC* inhibitor did not reduce *PVT1* expression, suggesting that *PVT1* expression is not controlled by *MYC*, but that both genes were regulated by *BRD4*. The reduction of *MYC* mRNA by *PVT1* knockdown indicated that *PVT1* regulates *MYC* expression at the transcriptional level.

Our observation of the high expression of *PVT1* and *MYC* in MM cells compared with normal plasma cells is consistent with those of previous studies [17]. Many studies have illustrated that *PVT1* levels are higher in many cancers with an increased *MYC* expression than in normal tissues and are associated with a poor prognosis [16,23,24,25]. Tissue microarray analysis of primary tumors indicated a high correlation between *PVT1* and *MYC* expression, providing strong evidence for cooperation between *PVT1* and *MYC* in different human cancers [29]. Nagoshi et al. reported that frequent *PVT1* rearrangements and the novel chimeric genes *PVT1-NBEA* and *PVT1-WWOX* occur in MM with 8q24 abnormality [30]. They also found a high expression of *PVT1* and *MYC* in most MM cell lines, regardless of *PVT1* or *MYC* rearrangement status. Our findings of *PVT1* elevation in plasma cells of MM compared to MGUS suggest a role of *PVT1* in the pathogenesis of malignant clonal plasma cells and the progression from precancerous stages to cancer.

The precise mechanisms controlling the expression of the lncRNA *PVT1* in MM have not been fully clarified. Increases in *MYC* copy number and *PVT1* expression occur in more than 98% of cancer cases with increased 8q24 copy numbers, indicating an interaction between *PVT1* and *MYC* and suggesting that they are part of a common signaling pathway. We could not find any correlation between *PVT-1* expression level and chromosomal abnormalities, including t(8;14) and 8q24 amplification detected using iFISH. This finding is not conclusive because iFISH is not sufficient for the detection of gene amplification; however, a positive correlation between *PVT1* and *MYC* expression implies a common system of regulation for both genes and indicates that upregulation of *PVT1* and *MYC* expression does not solely depend on 8q24 abnormalities in MM.

We speculated that a BRD4 inhibitor would downregulate the expression of *PVT1* and *MYC*. JQ1 and CPI -203 significantly reduced the expression of *PVT1* in myeloma cell lines. Combined with the high expression of *MYC* and *PVT1*, and the positive correlation between *MYC* and *PVT1* in the MM and MGUS patients’ plasma cells, these results suggest that *MYC* and *PVT1* are regulated by the same mechanism. There were two possible regulation mechanisms for *MYC* and *PVT1* expression. One is that *MYC* regulates *PVT1* expression and the other is that both *MYC* and *PVT1* are controlled by the super-enhancer. To test these hypotheses, we examined *PVT1* expression using 10058-F4, which suppresses the transcriptional activity of *MYC* by inhibiting the formation of an MYC and MAX heterodimer [31]. The administration of 10058-F4 did not decrease *PVT1* expression, except in the OPM2 cell line. These results suggest that *PVT1* is not transcriptionally regulated by MYC and that the two genes are simultaneously regulated by super-enhancers. This result is inconsistent with previous reports that the *PVT1* promoter region contains two enhancer E-boxes that serve as *MYC* binding sites and that E-box 2 mediates the binding of MYC to the *PVT1* promoter to promote *PVT1* expression [32]. We cannot fully explain this discrepancy. It may be caused by differences in the cell lineages or by differences in the conditions of the cells.

Many studies have revealed a relationship between *PVT1* and *MYC* expression. Our experiments with *PVT1* knockdown using the locked nucleotide antisense oligonucleotide resulted in decreased expression of *MYC* mRNA, a consistent observation with a previous report depicting the effect of *PVT1* on MYC transcription [33], but inconsistent with another report in which low levels of c-Myc protein were found without significant changes in *MYC* mRNA levels when siRNA was used to knock down *PVT1* [29].

We found a tendency toward shorter OS and PFS in the high *PVT1* group, although these differences did not reach statistical significance. These results suggested that a high level of *PVT1* might be associated with a prognosis of relapse and drug resistance. An association between high *PVT1* expression and poor prognosis was reported in several cancers [16,23,25]. The involvement of *PVT1* in resistance to anticancer drugs through several pathways was documented [24,34]. For example, *PVT1* acts as a competitive endogenous RNA that forms a tight network with protein-coding mRNAs, such as *CDH1, TP73, TP31, RUNX1*, and *RUNX* via microRNA-200, thereby regulating breast cancer progression [35,36]. For further clarification, we performed a subgroup analysis. The prognostic significance of high *PVT1* was apparent when the survival data excluding the patients with del 17p were analyzed. Del 17p is one of the worst prognostic factors; therefore, it strongly impacts survival and may mask the effects of *PVT1*. Research into treatment targeting *PVT1*, such as the use of BRD4 inhibitors, may be warranted only for some specific subcategories of patients. We acknowledge the argument that this type of subgroup analysis in a small number of patients cannot be used to draw definitive conclusions. However, our results provide some basis for further studies aimed at elucidating *PVT1* involvement in MM progression and drug resistance.

## 4. Materials and Methods

### 4.1. Cell Lines

The cell lines used are listed in Appendix A, with cytogenetic information. The human myeloma cell line RPMI8226 was obtained from the American Type Culture Collection (Rockville, MD, USA), and lines KMS11, KMS12PE, KMS12BM, KMM1, and KMS26 were kindly provided by Takemi Otsuki (Kawasaki Medical School, Okayama, Japan). OPM2 was kindly provided by Masaki Ri (Nagaya City University, Nagoya, Japan). All lines were cultured in RPMI 1640 medium (Sigma-Aldrich, St. Louis, MO, USA), supplemented with 10% fetal bovine serum at 37 °C and 5% CO_2_.

### 4.2. Patients

In this study, we included a total of 137 MM patients (including 96 newly diagnosed MM (NDMM), 26 relapse refractory MM (RRMM), and 15 smoldering MM (SMM)), 62 with MGUS, and 21 control patients with lymphoma without bone marrow infiltration or acute myeloid leukemia in complete remission. Patients were recruited from July 2010 to March 2016. The patient characteristics are summarized in Appendix A. This study was approved by Gunma University Hospital Clinical Research Review Board under the guidelines of the Declaration of Helsinki (project code 1295, approved data 22-Apr-2020). Bone marrow (BM) aspirate samples were obtained upon diagnosis after obtaining each patient’s informed consent.

### 4.3. Treatment with Inhibitors

Myeloma cell lines KMS11, KMS12PE, KMS12BM, KMS26, KMM1, OPM2, and RPMI8226 were treated with 1 μM JQ1 and 0.1 μM CPI203. The cell lines KMS11, KMS12PE, KMM1, OPM2, and RPMI8226 were treated for 24 h with an *MYC* inhibitor 10058-F4 (50 or 100 μM). Cell growth was determined using the WST-8 assay (Dojindo Laboratory, Kumamoto, Japan) at 24, 48, and 72 h. RNA was isolated from cells incubated for 24 h, and gene expression was determined using real-time PCR. The experiments were performed in triplicate.

### 4.4. PVT1 Silencing in Myeloma Cell Lines

RNase H-activating locked nucleic acid (LNA) GapmeR™ (Exiqon, Vedbaek, Denmark) was used to silence *PVT1* expression in vitro. The cell lines KMS12PE and OPM2 were cultured with GapmeR for 24, 48, and 72 h, after which cell viability was determined using the WST-8 assay. All experiments were performed in duplicate. Gene expression was determined after 72 h of treatment.

### 4.5. Isolation of Nucleic Acids

Plasma cells were purified from bone marrow mononuclear cells with anti-CD138 antibody conjugated with phycoerythrin (PE) (Beckman Coulter, Brea, CA, USA) and the Easy Step PE positive selection kits containing anti-PE antibodies conjugated with micro-magnetic beads (STEMCELL Technologies, Vancouver, BC, Canada). RNA was extracted from the plasma cells (and one autopsied extramedullary plasmacytoma of the liver) and cell lines using mirVana RNA Isolation kits (Ambion, Austin, TX, USA). Complimentary DNA (cDNA) was produced using PrimeScript™ RT reagent kits with gDNA Eraser (TaKaRa Bio, Kyoto, Japan).

### 4.6. Real-Time PCR Analysis of PVT1 Expression

The transcript levels, including those of PVT1, were determined using real-time PCR with the Power SYBR Green PCR Master Mix (Applied Biosystems, Foster City, CA, USA). Primers used for detection were as follows: PVT1: F-5′-CACTCTGGACGACTTGAGAAC-3′, R- 5′-TCCTCAGATGAACCAGGTGAACA-3′; MYC: F-5′-CCTGGTGCTCCATGAGGAGA-3′, R-5′-CAGTGGGCTGTGAGGAGGGTTT-3′; ACTB: F- 5′-TGGCACCCAGCAATGAA-3′, R- 5′-CTAAGTCATAGTCCGCCTAGAAGCA-3′. The expression levels were calculated using the ΔΔCt method. ACTB was used as an internal control, and cDNA extracted from the HL60 acute myeloid leukemia cell line was used as a calibration sample. The relative RNA expression levels are expressed as 2^−ΔΔ*C*t^.

### 4.7. Statistical Analysis

EZR version 1.41 (Saitama, Japan) was used for the statistical analysis. *p* values < 0.05 were considered significant. The frequencies were evaluated using Fisher’s exact tests, and the continuous values were evaluated using Mann–Whitney U tests or Kruskal–Wallis tests. The overall survival (OS) and the progression-free survival (PFS) were evaluated using the Kaplan–Meier method and log-rank test for univariate analysis. The Cox regression hazard model was used for multivariate analysis.

## 5. Conclusions

We observed a positive correlation between *MYC* and *PVT1* in patients, synchronous downregulation of *MYC* and *PVT1* by JQ1 and CPI203, and no effect of *MYC* inhibitor for *PVT1* expression, suggesting that the expression of these two genes is co-regulated by the *BRD4* complex. *PVT1* and *MYC* cooperation may contribute to MM pathogenesis and progression. Our results support the rationale for targeting *BRD4* in these patients.

## Figures and Tables

**Figure 1 ijms-21-07121-f001:**
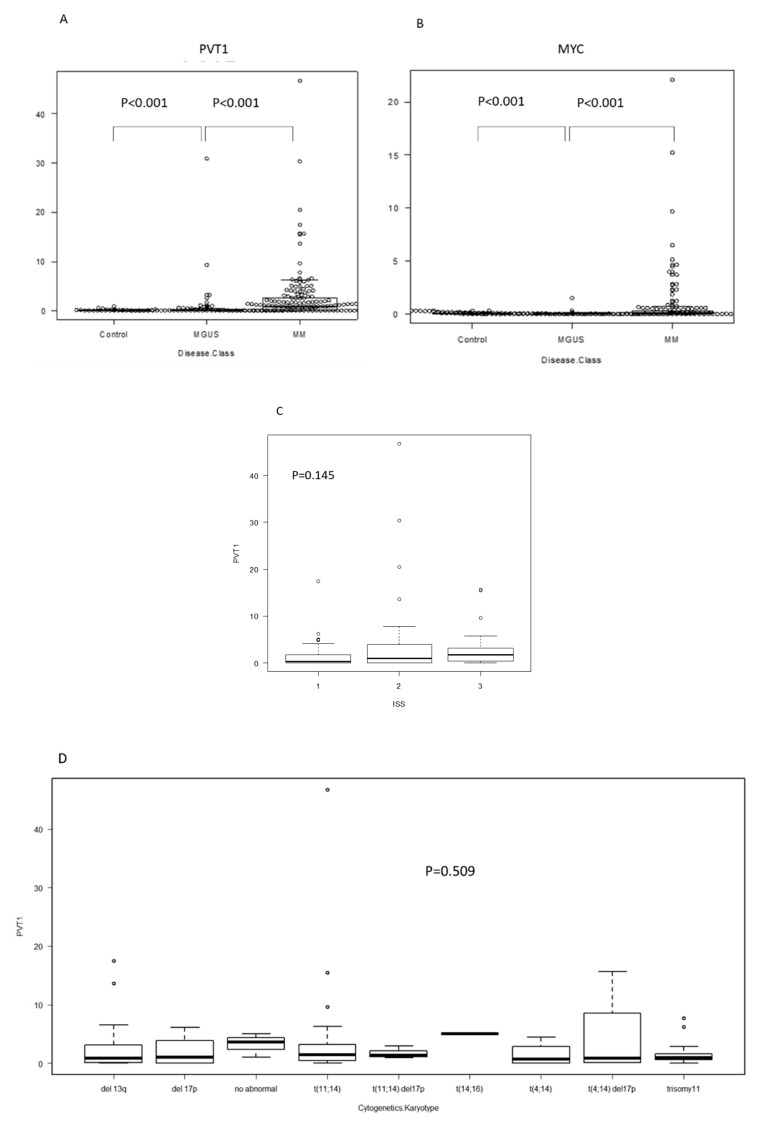
RNA expression determined using RQ-PCR in plasma cells isolated from bone marrow specimens, organized by patient status. Each dot represents a patient. RQ-PCR: real time quantitative PCR. MGUS: monoclonal gammopathy of undetermined significance. MM: multiple myeloma. (**A**) Plasmacytoma Variant Transcript 1 (PVT1), (**B**) MYC, and (**C**) PVT1 expression by international staging system (ISS). (**D**) PVT1 expression, according to karyotype, determined using fluorescence in situ hybridization (FISH). (**E**) PVT1 expression by chromosome 8q24 abnormality determined using FISH. (**F**) Correlation between PVT1 and MYC RNA expression in plasma cells of MM and MGUS.

**Figure 2 ijms-21-07121-f002:**
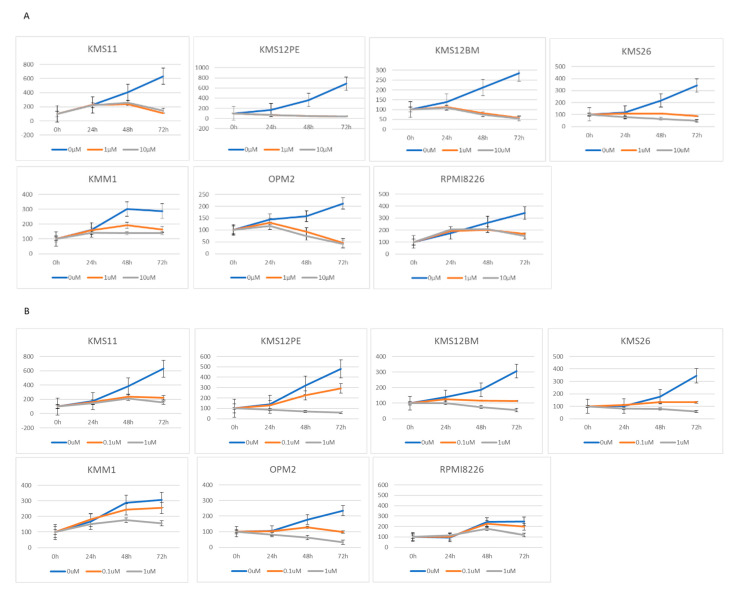
Growth determined using the WST-8 assay, and RNA expression determined using RQ-PCR in seven MM cell lines treated with two BRD4 inhibitors: JQ1 (0, 1, 10 μM) and CPI-203 (0, 0.1, 1 μM). (**A**) Cell growth under treatment with JQ1. (**B**) Cell growth under treatment with CPI-203. (**C**) *PVT1* expression under treatment with JQ1. (**D**) *PVT1* expression under treatment with CPI-203. (**E**) *MYC* expression under treatment with JQ1. (**F**) *MYC* expression under treatment with CPI-203.

**Figure 3 ijms-21-07121-f003:**
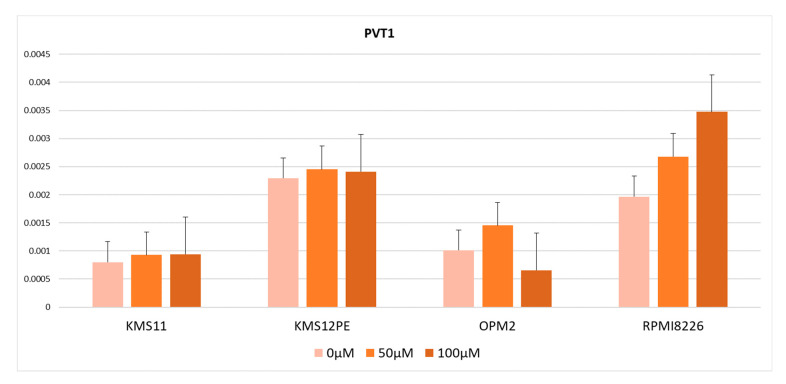
*PVT1* expression in cells treated with the *MYC* inhibitor 10058-F4. Error bars show the standard error of mean (SEM).

**Figure 4 ijms-21-07121-f004:**
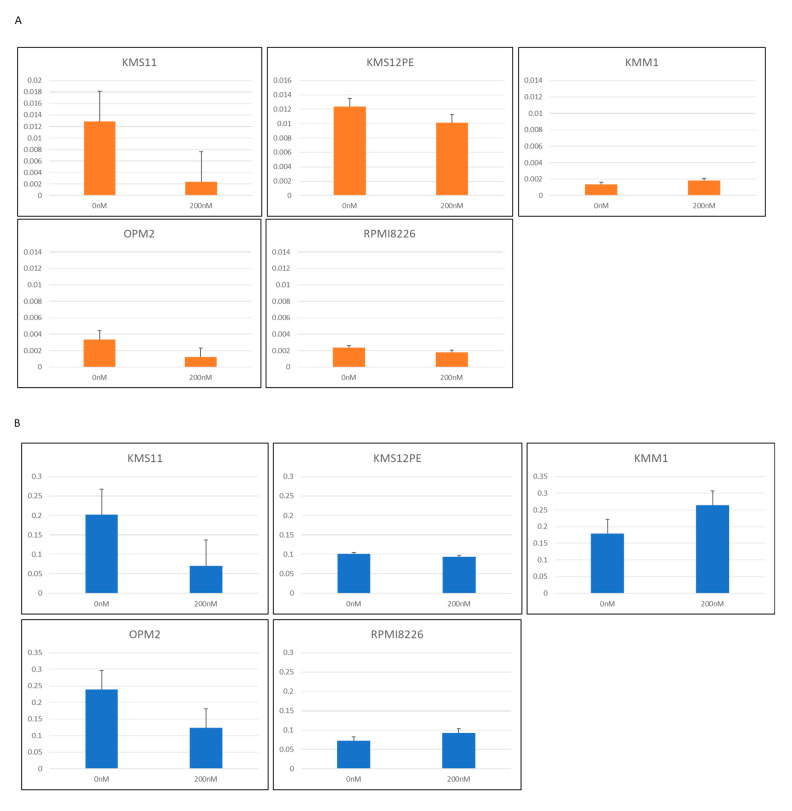
(**A**) *PVT1* expression in MM cell lines treated with antisense oligonucleotide LNA™ long RNA GapmeR for *PVT1*. (**B**) *MYC* expression in MM cell lines treated with antisense nucleotide LNA™ long RNA GapmeR for *PVT1*.

**Figure 5 ijms-21-07121-f005:**
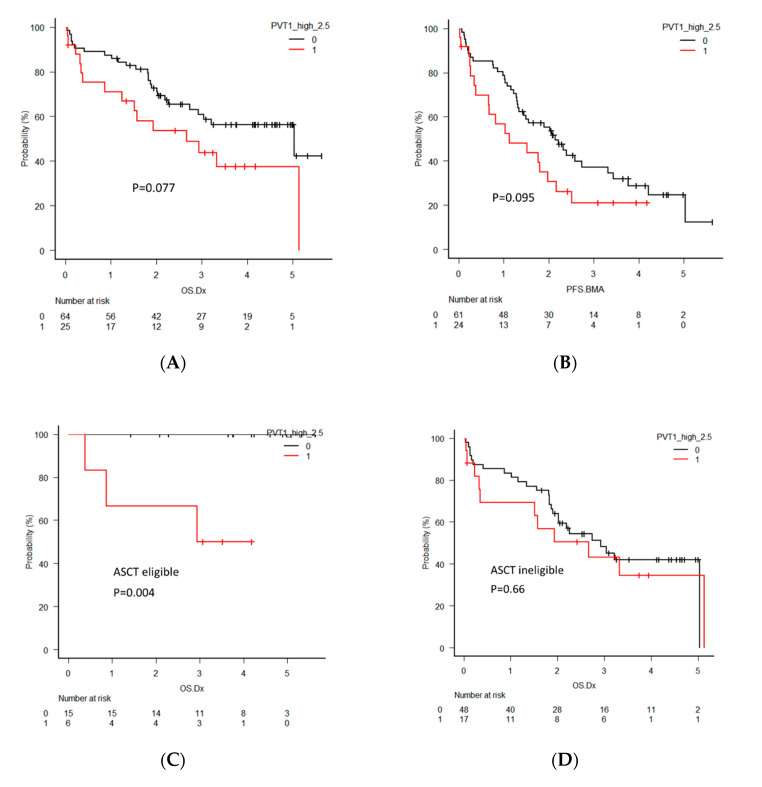
(**A**) Overall survival (OS) and (**B**) progression-free survival (PFS) in all newly diagnosed multiple myeloma (NDMM) patients divided into two groups by *PVT1* expression level. (**C**) OS in NDMM patients eligible for autologous stem cell transplantation (ASCT). (**D**) OS in NDMM ineligible for ASCT. (**E**) PFS in NDMM eligible for ASCT. (**F**) PFS in NDMM ineligible for ASCT. (**G**) OS in NDMM with high-risk cytogenetics. (**H**) OS in NDMM with standard-risk cytogenetics. (**I**) OS in NDMM with deletion 17p (del 17p). (**J**) OS in NDMM without del 17p. (**K**) PFS in NDMM with high-risk cytogenetics. (**L**) PFS in NDMM with standard-risk cytogenetics. (**M**) PFS in NDMM with del 17p. (**N**) PFS in NDMM without del 17p.

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
