# Peer review of "Long Noncoding RNA PVT1 Is Regulated by Bromodomain Protein BRD4 in Multiple Myeloma and Is Associated with Disease Progression"

_ijms, 2020, doi:10.3390/ijms21197121_

Round 1
Reviewer 1 Report
This is a detailed work on the roles of PVT1 and MYC in MM. However, it needs further explanation.
For OS and PFS, less than half of the MM patients (i.e,140) were evaluated (67 and 62 for OS and PFS analyses, respectively). Therefore, it can be difficult to draw solid conclusions about the roles of PVT1 and MYC collaboration in OS and PFS.
Author Response
This is a detailed work on the roles of PVT1 and MYC in MM. However, it needs further explanation.
For OS and PFS, less than half of the MM patients (i.e,140) were evaluated (67 and 62 for OS and PFS analyses, respectively). Therefore, it can be difficult to draw solid conclusions about the roles of PVT1 and MYC collaboration in OS and PFS.
Thank you for your valuable comments.
We have reviewed the patient records and collected further information to increase the patient number.
First, we would like to apologize for excluding three patients because these patients’ samples were collected at the time of remission and are thus inappropriate. Therefore, the total number of MM patients was 137.
Among those 137 cases, 96 cases were of newly diagnosed MM (NDMM), 26 were of relapse refractory MM (RRMM), and 15 were of smoldering MM (SMM).
We increased the clinical data for survival up to 89 for OS and 85 for PFS among the 96 NDMM patients.
We have added the details regarding the increase in the number of patients in the Material and Methods and Results sections.
Materials and Methods.
A total of 137 MM patients (including 96 newly diagnosed MM (NDMM), 26 relapse refractory MM (RRMM), and 15 smoldering MM (SMM)),
Results
Eighty-nine NDMM patients were analyzed for OS, and 85 NDMM patients were analyzed for PFS. Seven patients were excluded from OS analysis and 11 patients were excluded from PFS analysis due to lack of reliable clinical data.
Reviewer 2 Report
This is a very interesting study. Presentation is excellent.
Several queries and minor points should be addressed:
- Gold standard method for detection of acquired cytogenetic aberrations in plasma cell dyscrasias is interphase FISH (iFISH). If this was the method used it should be abbreviated throughout as iFISH, rather than FISH.
- Check throughout that abbreviations are annotated following first use of the full term, and then abbreviations used subsequently. For example, fluorescence in situ hybridisation followed by (FISH) (?iFISH) occurs in the Discussion, despite use of the abbreviation in earlier text.
- It would be good practice in the study (in fact in all cell line studies!) to check the reported acquired cytogenetic aberrations for the cell lines used. Further acquired aberrations may occur over time in cell lines and this may confuse interpretation of results in otherwise well-designed studies. Was this done?
- A simple table listing the cell lines used, and their reported acquired cytogenetic aberrations, would be a useful reference point for those reading the manuscript, providing it was within editorially defined limits for the numbers of tables and/or figures.
- Nomenclature for genes should be annotated in italics. Gene products should be annotated in normal font. Please check that this is adhered to throughout.
- When reading the text and the legend for Figure 1 it took me some time to understand it until I realised that Figure 1E came before 1D. It would be better if these could be inserted in alphabetical sequence.
- With regard to the outliers in several plots in Figure 1, have the authors any explanation for these?
- It would have been very interesting if the myeloma patient PFS and OS data in Figure 5 could be broken down into further subcategories. I recognise the main argument against doing this (ever smaller numbers in each subcategory) but grouping according to fairly general factors such as: high risk cytogenetics v standard/low risk cytogenetics; ASCT v non ASCT; drug regimens containing mAb v non mAb etc, whilst almost certainly not definitive, may suggest associations. This, in turn, may help inform clinical trial design in the context of trialling inhibition of BRD4
- With regard to the point above, do the authors have any information on the relative frequencies of high-risk cytogenetics, such as del17p, in the OS subgroups. ie could such aberrations account for shorter OS, either separately or in synergy with higher PVT1 expression, rather than higher PVT1 alone?
- Note I didn’t have access to any supplemental material.
Author Response
This is a very interesting study. Presentation is excellent.
Thank you for your encouraging comments. We really appreciate your special guidance and comments which have helped improve our work.
Several queries and minor points should be addressed:
- Gold standard method for detection of acquired cytogenetic aberrations in plasma cell dyscrasias is interphase FISH (iFISH). If this was the method used it should be abbreviated throughout as iFISH, rather than FISH.
Thank you for your guidance. We have changed FISH to iFISH.
- Check throughout that abbreviations are annotated following first use of the full term, and then abbreviations used subsequently. For example, fluorescence in situ hybridisation followed by (FISH) (?iFISH) occurs in the Discussion, despite use of the abbreviation in earlier text.
Thank you for your guidance. We have checked the abbreviations and revised them.
- It would be good practice in the study (in fact in all cell line studies!) to check the reported acquired cytogenetic aberrations for the cell lines used. Further acquired aberrations may occur over time in cell lines and this may confuse interpretation of results in otherwise well-designed studies. Was this done?
Thank you for your comments. We checked cytogenetic aberrations for the cell lines used. We completely agree with your comments that acquired aberrations may occur over time in cell lines and this may confuse interpretation of results; however, we apologize for conducting this study because of cost and time issues.
- A simple table listing the cell lines used, and their reported acquired cytogenetic aberrations, would be a useful reference point for those reading the manuscript, providing it was within editorially defined limits for the numbers of tables and/or figures.
Thank you for your valuable comments. We have added a table listing the cell lines used with their reported acquired cytogenetic aberrations and added supplementary table.
- Nomenclature for genes should be annotated in italics. Gene products should be annotated in normal font. Please check that this is adhered to throughout.
Thank you for your guidance. We checked the font and revised the text following your comments.
- When reading the text and the legend for Figure 1 it took me some time to understand it until I realised that Figure 1E came before 1D. It would be better if these could be inserted in alphabetical sequence.
I am very sorry for the confusing text. I have inserted Figure1D and 1E in alphabetical sequence.
- With regard to the outliers in several plots in Figure 1, have the authors any explanation for these?
Thank you for your valuable question. We checked the patient data with outliers. The cases included smoldering MM (SMM), relapse refractory MM (RRMM), and newly diagnosed MM (NDMM); we did not observe any common clinical features of the patients. However, many patients with NDMM progressed or died relatively earlier. Regarding relevance with cytogenetics, those outliers were not associated with high risk and were rather associated with t(11;14) or del 13q.
- It would have been very interesting if the myeloma patient PFS and OS data in Figure 5 could be broken down into further subcategories. I recognise the main argument against doing this (ever smaller numbers in each subcategory) but grouping according to fairly general factors such as: high risk cytogenetics v standard/low risk cytogenetics; ASCT v non ASCT; drug regimens containing mAb v non mAb etc, whilst almost certainly not definitive, may suggest associations. This, in turn, may help inform clinical trial design in the context of trialling inhibition of BRD4
Thank you very much for your special and worthful guidance. We have broken the data down into subcategories following your guidance. We also found the clinical significances of the ASCT group and the no del 17p group. We have improved Figure 5 (A through N) and revised the results and discussion as follows.
To investigate the clinical significance of PVT1 in MM, the overall survival (OS) and progression free survival (PFS) of MM patients were analyzed by dividing the two groups according to PVT1 expression levels using the quartile method. Eighty-nine NDMM patients were analyzed for OS, and 85 NDMM patients were analyzed for PFS. Seven patients were excluded from OS analysis, and 11 patients were excluded from PFS analysis due to lack of reliable clinical data. The patients with high PVT1 expression tended to have a shorter OS with a median of 2.7 years compared to 5.0 years in those with low expression (p = 0.077) (Figure 5A). PFS also tended to be shorter in patients with high PVT1 expression, but did not reach statistical significance (p = 0.095) (Figure 5B).
Results section
For further clarification of the clinical significance of PVT1 expression, OS and PFS were analyzed separatory by subcategories, including patients eligible or ineligible for autologous stem cell transplantation (ASCT), harboring high risk or standard risk cytogenetics, and harboring deletion 17p (del 17p). The ASCT eligible patients with high PVT1 expression showed significantly shorter OS (p=0.004) but such a significant difference was not found in the ASCT ineligible patients (p=0.66). PFS in the patients with high PVT1 expression in both groups was not different. OS and PFS between patients with high and low PVT1 expression did not differ in the groups of patients harboring high risk and standard risk cytogenetics. However, significantly shorter OS and PFS of the patients with high PVT1 expression was found in the groups of patients who did not harbor del 17p (p=0.008, p=0.0043) although such a significant difference was not observed in the patients with del 17p.
Discussion
For further clarification, we performed subgroup analysis. The prognostic significance of high PVT1 was apparent when the survival data excluding the patients with del 17p were analyzed. Del 17p is one of the worst prognostic factors, therefore it strongly impacts survival, and may mask the effects of PVT1. Research into treatment targeting PVT1, such as the use of BRD4 inhibitors, may be warranted only for some specific subcategories of the patients. We acknowledge the argument that this type of subgroup analysis in a small number of patients cannot be used to draw definitive conclusions. However, our results provided some basis for further studies aimed at elucidating PVT1 involvement in MM progression and drug resistance.
- With regard to the point above, do the authors have any information on the relative frequencies of high-risk cytogenetics, such as del17p, in the OS subgroups. ie could such aberrations account for shorter OS, either separately or in synergy with higher PVT1 expression, rather than higher PVT1 alone?
Thank you for your comments. As described above, we reanalyzed the data following your suggestion and found that high PVT1 affects the survival only in patients without del 17p. No synergy was observed, as you expected. The relative frequencies of high-risk cytogenetics and del 17p have been added to the patient characteristics table (supplementary table 2)
- Note I didn’t have access to any supplemental material.
We apologize for the issue. We have reuploaded the tables.